# BLESS: Benchmarking Large Language Models on Sentence Simplification

**Tannon Kew**[1,†], **Alison Chi**[2,†], **Laura Vásquez-Rodríguez**[3,4,†,*],
**Sweta Agrawal**[5], **Dennis Aumiller**[6], **Fernando Alva-Manchego**[7], **Matthew Shardlow**[8]

[1]University of Zurich, Switzerland [2]National Tsing Hua University, Taiwan
[3]Idiap Research Institute, Switzerland [4]University of Manchester, UK
[5]University of Maryland, US [6]Cohere, US [7]Cardiff University, UK
[8]Manchester Metropolitan University, UK

kew@cl.uzh.ch, achi@gapp.nthu.edu.tw, laura.vasquez@idiap.ch
sweagraw@umd.edu, dennisaumiller@cohere.com
alvamanchegof@cardiff.ac.uk, m.shardlow@mmu.ac.uk

## Abstract

We present **BLESS**, a comprehensive performance benchmark of the most recent state-of-the-art large language models (LLMs) on the task of text simplification (TS). We examine how well off-the-shelf LLMs can solve this challenging task, assessing a total of 44 models, differing in size, architecture, pre-training methods, and accessibility, on three test sets from different domains (Wikipedia, news, and medical) under a few-shot setting. Our analysis considers a suite of automatic metrics as well as a large-scale quantitative investigation into the types of common edit operations performed by the different models. Furthermore, we perform a manual qualitative analysis on a subset of model outputs to better gauge the quality of the generated simplifications. Our evaluation indicates that the best LLMs, despite not being trained on TS, perform comparably with state-of-the-art TS baselines. Additionally, we find that certain LLMs demonstrate a greater range and diversity of edit operations. Our performance benchmark will be available as a resource for the development of future TS methods and evaluation metrics.[1]

## 1 Introduction

Large pre-trained language models (LLMs) have demonstrated strong performance on a wide range of NLP tasks without the need for task-specific fine-tuning, leading to a prevailing conventional wisdom that LLMs can solve *any* task. This has motivated the development of benchmarks to better understand the abilities of LLMs in specific domains such as healthcare (Sallam, 2023), finance (Dowling and Lucey, 2023), education (Baidoo-Anu and Owusu Ansah, 2023), engineering (Soba-

nia et al., 2023), and ethics (Zhuo et al., 2023), as well as for specific NLP tasks (Li et al., 2022; Wang et al., 2023; Liu et al., 2023).

However, it remains unclear how well current LLMs can perform on the challenging task of text simplification (TS). In this paper, we focus on sentence simplification in English, which typically involves rephrasing part or all of a sentence into language which is more accessible and easier to understand. While recent work has focused on evaluating TS abilities of select models, such as GPT-3.5-Turbo (Feng et al., 2023) and mT5 (Ryan et al., 2023), there is currently no large-scale and detailed analysis of the simplification capabilities of different LLMs.

In this study, we expand both the breadth and depth of the knowledge base on TS with LLMs, evaluating a wider variety of models on three different TS datasets: ASSET (Alva-Manchego et al., 2020a), NEWSELA (Jiang et al., 2020) and MED-EASI (Basu et al., 2023). We select these datasets to cover a variety of domains (Wikipedia, news, and medical) and a diverse set of TS operations (e.g. paraphrasing, splitting, and elaboration).

Specifically, we use in-context learning (ICL) and assess LLMs in a few-shot setting, experimenting with three different prompts. We select 44 widely used generative models (both open and closed-weight) and evaluate their abilities from three distinct angles. First, we rely on automatic evaluation metrics commonly used in the TS literature. Second, we quantify and compare the edit operations performed by the LLMs during simplification. Finally, we perform a targeted qualitative analysis to validate our findings and to better understand the quality of the generated simplifications. Our findings reveal that closed-weight models provide significant gains over open-weight alternatives under a few-shot setting, establishing them as a

---

[1]We make our code and the generated system outputs available at https://github.com/ZurichNLP/BLESS.

[†]These authors contributed equally.

[*]Work done as a PhD student at the University of Manchester, United Kingdom.

strong baseline for future work on TS. We summarize our contributions as follows:

1. BLESS (**B**enchmarking **L**arge language mod**E**ls on **S**entence **S**implification), a performance evaluation benchmark of 44 LLMs in a few-shot setting (Section 3).

2. An evaluation that includes both widely used automatic metrics and an analysis of the TS edit operations performed by the models (Section 4).

3. A qualitative analysis of the results, with manual annotation of simplification operations and an examination of the relationships between selected evaluation metrics (Section 5).

## 2 Related Work

**Text Simplification Benchmarks**  Most simplification work treats the task as a monolingual machine translation problem, training models on datasets containing complex-simple sentence pairs (Zhu et al., 2010). Alva-Manchego et al. (2020b) performed a standardized evaluation of general data-driven simplification systems, using Wikipedia-based datasets and NEWSELA. At the document level, Alva-Manchego et al. (2019b) conducted a systematic analysis of simplification operations to demonstrate the limitations and disruptions that occur when multiple sentences are involved. Benchmarks have also been established for more specific kinds of simplification: for example, both non-neural (Paetzold and Specia, 2016) and neural (Stajner et al., 2022; Saggion et al., 2022) approaches to lexical simplification, which aims to replace complex words with simpler alternatives.

**LLM-based Simplification**  LLMs such as `GPT-3.5-Turbo`, the model behind early versions of ChatGPT[2], are often used out-of-the-box without any further training for a given domain or task. Some previous works have investigated simplification capabilities of select LLMs in order to benchmark performance against dedicated approaches (Aumiller and Gertz, 2022; Vásquez-Rodríguez et al., 2022; Ryan et al., 2023; Sun et al., 2023; Chi et al., 2023). Meanwhile, Feng et al. (2023) explored the TS abilities of the two strong-performing OpenAI models, `GPT-3.5-Turbo` and `Davinci-003`. However, despite these efforts, we only have results from a very limited number of LLMs and evaluation metrics. Thus, it remains un-

---

[2]https://chat.openai.com/

| Dataset | Domain | Size | # Words | | # R | TER |
|---|---|---|---|---|---|---|
| | | | C | S | | |
| ASSET | Wikipedia | 359 | 22.57 | 18.87 | 10 | 16.79 |
| MED-EASI | Medical | 300 | 26.48 | 27.42 | 1 | 25.03 |
| NEWSELA | News | 256 | 26.44 | 24.82 | 4 | 23.17 |

Table 1: Dataset Statistics. C: Complex; S: Simple; R: References. TER refers to Translation Error Rate, a measurement of the average edit distance between the source and reference texts (see https://www.cs.umd.edu/~snover/tercom).

clear how a wider spectrum of models, differing in architecture and training strategy, perform on different domains and in response to different prompts. We aim to fill this gap and study the simplification abilities of 44 LLMs in order to highlight potential weaknesses and determine areas for further development. To the best of our knowledge, we are the first to focus on establishing the performance of recent LLMs on the task of TS.

## 3 BLESS: Benchmarking Large Language Models on Sentence Simplification

### 3.1 Datasets

Our assessment establishes the performance of current LLMs on TS according to three datasets, covering different domains and styles. Table 1 summarizes these datasets.

**ASSET** (Alva-Manchego et al., 2020a) comprises 2,359 sentences from English Wikipedia paired with 10 simplified references. We use the official test split (359 sentences) for evaluation. These references were created by crowdworkers who were instructed to use edit operations such as replacement, splitting, and deletion.

**MED-EASI** (Basu et al., 2023) is a simplification dataset for short medical texts containing 1,979 complex (expert) - simple (layman) pairs. Each text contains one or more sentences. In this dataset, simplified texts are composed using four types of operations: elaboration, replacement, deletion, and insertion. We use the released test split (300 instances) for our evaluation. Unlike the other two datasets, simplifications in MED-EASI are slightly longer than the complex source texts, due to explanation and decomposition of complex medical terms.

**NEWSELA** (Xu et al., 2015) contains 1,130 long-form news articles that have been professionally

rewritten according to four different graded readability levels. For our benchmarking experiments, we opt for the Newsela-Manual test set (Jiang et al., 2020). We extract all aligned and partially aligned sentence pairs between a complex source sentence (level 0) and the four simplified article versions (levels 1-4), keeping only those sentences for which we have a reference for all four simplification levels.[3] This results in 256 test examples. Using this small subset of NEWSELA data ensures that sentence-level alignments are of high quality and capture important edit operations such as splitting.

## 3.2 LLM Types

We investigate a total of 44 LLMs with different sizes, architectures, and training objectives. The models we consider range from 60 million to 176 billion parameters and are all based on the transformer architecture (Vaswani et al., 2017), consisting of either an encoder-decoder or a standalone decoder. Furthermore, all have undergone a self-supervised pre-training stage. Nine of these models leverage instruction-tuning, which fine-tunes a pre-trained base model on labeled instruction-response pairs from a diverse set of tasks. Finally, just three of these models have received additional training through reinforcement learning with human feedback (RLHF) to better align the model's responses with human preferences (Stiennon et al., 2020; Ouyang et al., 2022). Evaluating a wide variety of currently available models should serve as a broad baseline and give sufficient information on which models perform best in which domains as well as where key challenges remain.

We broadly distinguish between open- and closed-weight models. The former pertains to models for which the trained weights are accessible and thus allow for self-hosting. Typically, these models are considered to be "open-source." However, we note that this obfuscates specific licensing agreements attached to some models and whether or not the training data and code are also made available. In comparison, closed-weight models refer to those whose weights are kept private and can be queried only through APIs. Our open-weight models include variants of the T5 family (Raffel et al., 2020), GPT-style models (Radford et al., 2019; Wang and Komatsuzaki, 2021), OPT (Zhang et al., 2022c)

and LLaMA models (Touvron et al., 2023), and the BLOOM family (Scao et al., 2022). For closed-weight models, we focus on those developed by OpenAI. Details on each model family are provided in Appendix A.

## 3.3 Prompts

To simplify sentences with LLMs without additional fine-tuning, we use in-context learning (ICL). ICL is a prompting technique that utilizes a small number of input-output examples to demonstrate a task (Brown et al., 2020). Previous work on related tasks has demonstrated that LLMs are sensitive to which input prompts and few-shot examples are used (Zhang et al., 2022b; Lu et al., 2022; Agrawal et al., 2023). To account for this, we construct three stylistically distinct prompts that consist of a task instruction and $N$ few-shot examples (see Figure 1). For all generation settings, we set $N=3$ and randomly sample complex-source pairs from the corresponding validation sets. We leave a detailed investigation of optimal in-context learning strategies for TS to future work.

## 3.4 Inference Settings

For open-weight models, we run inference on local GPUs using the Transformers library (Wolf et al., 2020). We load the models with 8-bit quantization (Dettmers et al., 2022), which allows us to run inference efficiently on as few as 5 A100 80GB GPUs. For closed-weight models, we use the APIs provided by OpenAI. As generation hyperparameters, we use Nucleus Sampling (Holtzman et al., 2020) with a probability threshold of 0.9, a temperature of 1.0, and a maximum output length of 100 tokens. To account for the stochastic generation settings, we perform each inference run with 3 different random seeds and aggregate the results for each metric.

## 3.5 Baselines

We use the MUSS (Martin et al., 2022) model as our main baseline since it has been shown to achieve state-of-the-art performance. MUSS fine-tunes a BART-large (Lewis et al., 2020) model with ACCESS control tokens (Martin et al., 2020) extracted from labeled TS datasets and/or mined paraphrases to train both supervised (MUSS-wiki-mined) and unsupervised (MUSS-mined) TS systems. We use the suggested hyperparameters from the original paper to set the control tokens for simplification generation.

---

[3]These articles are simplified as a whole to match the desired school grade; therefore, there is no guarantee that there will be an exact match for all the sentences in the text across all grade levels.

(a) Prompt 0 uses a basic instruction adapted from (Feng et al., 2023) followed by a list of $N$ few-shot examples before the input sentence to be simplified.

(b) Prompt 1 uses the same basic task instruction as prompt 0, but presents few-shot examples in an inline, continuous text format.

(c) Prompt 2 repurposes the instructions from (Alva-Manchego et al., 2020a) that were provided to crowdworkers in the creation of the ASSET dataset. Similarly to prompt 0, few-shot examples are presented in a structured format.

Figure 1: Prompts used for LLM text simplification. The blue boxes contain the task instructions. Orange boxes show how the few-shot examples are presented to the model and yellow boxes contain the prefix for the model to continue.

## 3.6 Automatic Metrics

To assess how well LLMs can perform TS, we evaluate all the model outputs using a suite of automatic metrics.[4] We measure simplicity using SARI (Xu et al., 2016), meaning preservation using BERTScore (Zhang et al., 2020), and readability using FKGL (Kincaid et al., 1975). These metrics are computed using the EASSE package (Alva-Manchego et al., 2019a).[5] Additionally, we report LENS (Maddela et al., 2023), a recently proposed learned metric, which considers both the semantic similarity and the degree of simplification performed by the system with respect to the source sentence and references.[6] Where possible, we also establish the performance of 'gold' simplifications by evaluating available reference sentences using a 'leave-one-out' strategy. That is, in cases where multiple references are available, we select one at random and evaluate it against the remaining references.

## 4 Automatic Evaluation Results

In this section, we present the results of our automatic evaluation of simplification outputs and summarize our main findings. First, we perform an exhaustive assessment using automatic metrics (Section 3.6). For brevity, we report the results of the best-performing LLMs with SARI and BERTScore in Table 2 and provide the complete results for all 44 models and metrics in Appendix B. Then, we compute edit distance statistics to quantify the simplification operations performed by each of the LLMs (Section 4.1). We begin by assessing the impact of the different prompt formats.

**Structured prompting improves performance.**
Figure 2 reveals that prompts 0 and 2 both offer a slight advantage over prompt 1, especially in regard to meaning preservation. This confirms that providing a structured template for few-shot examples instead of embedding them within sentences is the most beneficial. Hence, we focus on prompt 2 for all our analysis, as it provides the most detailed description of the task and has also been used in prior work (Maddela et al., 2023).

**Training method matters more than size.** Table 2 presents the performance according to SARI

---

[4]See Appendix B.1 for details on each evaluation metric.
[5]https://github.com/feralvam/easse
[6]We compute LENS using its original implementation: https://github.com/Yao-Dou/LENS.

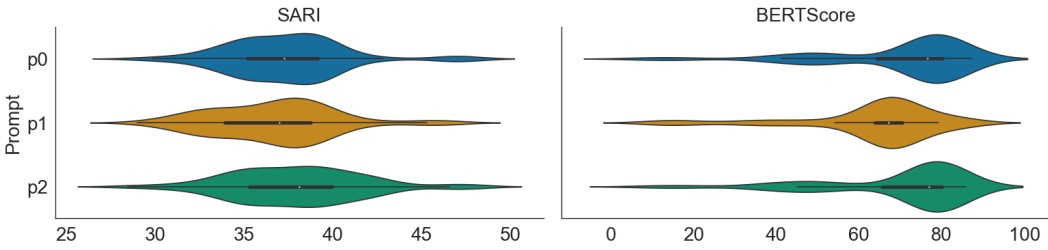

Figure 2: Impact of prompt selection on SARI and BERTScore for all models on ASSET. Prompts 0 and 2 achieve improved meaning preservation over prompt 1.

|  |  | ASSET | | MED-EASI | | NEWSELA | |
|---|---|---|---|---|---|---|---|
|  |  | SARI↑ | BERT↑ | SARI↑ | BERT↑ | SARI↑ | BERT↑ |
| **Baselines** | Gold References | 45.27 | 78.89 | 100 | 100 | 60.11 | 87.66 |
|  | MUSS-mined | 42.29 | 79.86 | 35.15 | 42.55 | 38.40 | 72.14 |
|  | MUSS-wiki-mined | 44.90 | 77.71 | 35.12 | 43.07 | 41.24 | 74.1 |
| **LLMs** | Ada-001* | 33.97 | 81.76 | 36.52 | 33.95 | 34.42 | 70.33 |
|  | Babbage-001* | 38.44 | 82.46 | 36.6 | 37.95 | 36.41 | 62.99 |
|  | Curie-001* | 39.87 | 82.75 | 38.22 | 39.31 | 37.53 | 69.17 |
|  | Davinci-002* | 42.84 | 85.91 | 36.34 | 43.67 | 40.25 | 73.62 |
|  | Davinci-003* | 46.60 | 79.66 | 39.81 | 40.83 | 37.76 | 61.56 |
|  | GPT-3.5-Turbo* | 47.69 | 79.39 | 40.14 | 40.67 | 37.29 | 60.19 |
|  | BLOOM | 39.72 | 76.63 | 37.72 | 11.95 | 37.48 | 61.17 |
|  | BLOOMZ | 37.63 | 82.06 | 36.6 | 12.9 | 37.06 | 69.55 |
|  | OPT-1.3b | 33.01 | 75.57 | 34 | 3.82 | 34.76 | 50.78 |
|  | OPT-30b | 38.04 | 77.22 | 35.08 | 9.96 | 37.58 | 61.79 |
|  | OPT-IML-MAX-1.3b | 36.00 | 79.73 | 37.01 | 11.85 | 37.08 | 62.68 |
|  | OPT-IML-MAX-30b | 42.03 | 79.39 | 35.8 | 11.73 | 39.59 | 66.39 |
|  | Flan-T5-small | 38.57 | 77.26 | 36.65 | 38.6 | 37.72 | 68.15 |
|  | Flan-T5-base | 41.40 | 79.7 | 36.79 | 40.63 | 38.67 | 68.09 |
|  | Flan-T5-large | 42.17 | 80.44 | 35.71 | 41.31 | 39.08 | 70.27 |
|  | Flan-T5-xl | 41.07 | 85.06 | 33.21 | 44.12 | 37.51 | 75.5 |
|  | Flan-T5-xxl | 41.75 | 84.13 | 34.27 | 43.43 | 39.42 | 73.05 |
|  | Flan-UL2 | 42.83 | 84.34 | 35.31 | 42.8 | 40.27 | 73.23 |

Table 2: For brevity, we report automatic metrics for simplification (SARI) and meaning preservation (BERTScore) for select models using Prompt 2. '*' indicates closed-weights. The full list of results is available in Tables 6, 7, and 8 in the Appendix.

and BERTScore for the top-performing LLMs. Scaling LLMs has revealed strong benefits in few-shot settings (Brown et al., 2020; Chowdhery et al., 2022); however, in our evaluation, we observe numerous exceptions to this rule. For example, Flan-T5-large (770 million parameters) consistently attains higher SARI scores on ASSET than Flan-T5-xl (3 billion parameters) and Flan-T5-xxl (11 billion parameters).[7] Meanwhile, we observe that training strategies such as

instruction-tuning and RLHF help to deliver greater improvements, especially for meaning preservation, as measured by BERTScore. This agrees with previous findings that demonstrate the benefits of instruction-based adaption strategies for improved generalization abilities (Schick and Schütze, 2021; Zhang et al., 2022a; Chung et al., 2022).

**ASSET** On Wikipedia-style data, OpenAI's Davinci-003 and GPT-3.5-Turbo outperform all other tested LLMs by a considerable margin according to SARI. Strikingly, these models also outperform the ground truth references, which are closely

---

[7]We include a wider comparison of selected LLMs on ASSET in Figure 7 in the Appendix.

approximated by the previous state-of-the-art `MUSS` models. This is notable since `MUSS-wiki-mined` was trained on the in-domain TS dataset of Wiki-Large (Zhang and Lapata, 2017). Meanwhile, for open-weight contenders, we can see in Table 2 that only a small number of models are competitive, namely `OPT-IML-Max-30b`, `Flan-T5-large`, and `Flan-UL2`, which scores the best balance between simplicity and meaning preservation according to automatic metrics.

**MED-EASI** For medical-related texts, we observe that the majority of the models consistently fail to preserve meaning (our qualitative analysis in Section 5 confirms this, see Table 3). The drop in meaning preservation can likely be explained by the fact that models are known to produce inadequate generations in out-of-domain settings (Müller et al., 2020; Singhal et al., 2023). The models that do strike a reasonable balance with both SARI and BERTScore are again OpenAI's more powerful offerings and the Flan models. Notably, we also observe that the two MUSS models are able to perform competitively with the Flan models despite being multiple orders of magnitude smaller.

**NEWSELA** Evaluating LLMs on professionally written simplifications from NEWSELA reveals that even the best LLMs are not able to match human performance. This is observable through the clear margins of around 20 SARI points and 14 BERTScore points between the best performers and the gold simplifications. On this dataset, `MUSS-wiki-mined` remains a strong baseline, outperforming all LLMs on both metrics, while `Davinci-002`, `Flan-UL2`, and `Flan-T5-xxl` show the strongest performances among the LLMs.

### 4.1 Analysis of Edit Operations

To identify the main token-level edit operations performed by LLMs, we use an adaptation of the Wagner–Fischer algorithm (Wagner and Fischer, 1974), following previous work by Vásquez-Rodríguez et al. (2021a). Specifically, we calculate the portion of insertion, replacement, deletion, and keep operations between the input source sentence and each of the system outputs for each dataset.

Figure 3 shows the distribution of token-level edit operations for the best-performing LLMs on ASSET (for a more comprehensive view across all datasets and models, see Figure 5 in the Appendix). Most models perform all four operations to differing degrees; however, similar to the gold

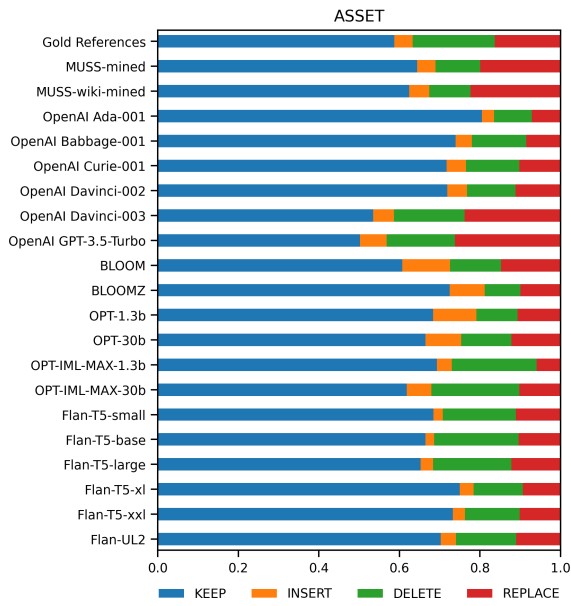

Figure 3: Distribution of token-level edit operations produced by the best-performing LLMs.

references, the keep operation is by far the most prominent in this dataset. Notably, `Davinci-003` and `GPT-3.5-Turbo` perform the most diverse set of operations, with fewer additions and more replacements than other models. Insertions are typically less frequent, suggesting that the majority of the models avoid adding new and potentially irrelevant content. We observe that most LLMs are within the range of the gold references in terms of the amount of information they delete when simplifying.

## 5 Qualitative Analysis

Automatic metrics are known to have blind spots and are not always entirely reliable (Alva-Manchego et al., 2021; He et al., 2022). To compensate for this, we perform a qualitative analysis on a total of 300 system outputs.

First, we check whether or not each output is a valid simplification and highlight common failure cases such as inappropriate changes to the meaning of the original text, ungrammatical outputs, and the occurrence of hallucinations. Then, we annotate occurrences of common simplification edit operations such as lexical simplification, deletion, sentence splitting, reordering, and paraphrasing.[8]

For our annotations, we select model outputs from the top five systems ranked according to per-

---

[8] All annotations were completed by one of the authors and validated separately by another.

| Model outputs | %S↑ | %MP↑ | %L+ | %P+ | %D+ | %Sp+ | %R+ | %H↓ |
|---|---|---|---|---|---|---|---|---|
| All | 61.67 | 67.33 | 30.33 | 28.33 | 35.0 | 4.33 | 4.67 | 12.33 |
| Top 5 SARI | 72.0 | 68.0 | **48.0** | **34.66** | 37.33 | **6.67** | 6.67 | 8.0 |
| Top 5 BERT | 62.67 | **84.0** | 17.33 | 29.33 | 34.67 | 5.33 | **10.67** | 2.67 |
| Top 5 FKGL | 34.67 | 40.0 | 14.66 | 17.33 | 26.67 | 0.0 | 0.0 | 36.0 |
| Top 5 LENS | **77.33** | 77.33 | 41.33 | 32.0 | **41.33** | 5.33 | 1.33 | **2.67** |
| Open-Weight | 58.58 | 64.55 | 29.47 | 22.76 | **36.94** | 3.36 | 3.73 | 13.81 |
| Closed-Weight | **87.50** | **90.63** | 37.50 | 75.0 | 18.75 | **12.50** | 12.50 | **0.0** |
| On ASSET | **77.0** | **82.0** | 31.0 | **54.0** | 33.0 | **8.0** | 4.0 | **10.0** |
| On NEWSELA | 54.0 | 70.0 | **34.0** | 9.0 | **38.0** | 5.0 | **7.0** | 17.0 |
| On MED-EASI | 54.0 | 50.0 | 26.0 | 22.0 | 34.0 | 0.0 | 3.0 | **10.0** |

Table 3: Results of our manual analysis. The annotation schema includes the following annotation features: S↑: accepted simplification, MP↑: meaning preserved, L+: lexical simplification, P+: paraphrasing, R+: reordering (no changes), D+: deletion, Sp+: sentence splitting, H↓: hallucination.

formance on the individual evaluation metrics of SARI, BERTScore, FKGL, and LENS. In each ranking set, we randomly select five complex-simple pairs from all generation settings. To evaluate a unique set of models for greater diversity, if a system is repeated in the ranking (e.g. two different prompt types from the same model appear in the top five), we choose the next best system for analysis. An example of our annotated outputs is shown in Table 9 in the Appendix. Table 3 shows results from this analysis, which we describe according to different criteria below.

**By Automatic Metric** Overall, we find that simplicity and meaning preservation are fairly balanced. However, there is a clear trade-off between these two axes when we consider the top 5 models according to SARI and BERTScore. This agrees with earlier findings from Schwarzer and Kauchak (2018). Along with a higher degree of simplicity, the top 5 SARI models exhibit more diverse edit operations than those ranked highly by BERTScore.

LENS, however, does not trade off simplicity and meaning preservation and even achieves a higher simplicity score than SARI along with its increased level of deletion. This result is in line with the previous finding that LENS achieves stronger correlations with human judgments compared to existing TS metrics (Maddela et al., 2023). The top 5 models ranked by FKGL, on the other hand, produce outputs with low simplicity and meaning preservation and an especially high amount of hallucinations. This result supports the previous finding that FKGL can be easily gamed by degenerations (Tanprasert and Kauchak, 2021) and is therefore an unsuitable metric for evaluating the outputs of automatic TS systems.

**By Open-Status** Open-weight models most frequently use the operations of lexical simplification, paraphrasing, and deletion, while structural operations such as sentence splitting and reordering are often neglected. Many only achieve high meaning preservation by directly copying the input sentence. However, the closed-weight models investigated here behave very differently: they produce close to 10% more splitting, lexical simplification, and re-ordering than open-weight ones, while simultaneously performing fewer deletions. This leads to a greater degree of paraphrasing.

**By Domain** When comparing performance between different domains, we observe that all LLMs do significantly better on general encyclopedic texts in ASSET in terms of both simplicity and meaning preservation, while also exhibiting a diverse set of edit operations. Although outputs from NEWSELA contain more hallucinations, meaning preservation is still fairly high. Outputs from MED-EASI, on the other hand, have the lowest meaning preservation by far and the least diverse set of edit operations. We find that MED-EASI outputs, along with others that do not preserve meaning, often contain repetitions, hallucinations, and in some cases even copy the input prompt, demonstrating a tendency to disregard the instruction and thus fail to complete the task. These failure modes are most frequently observed from the smaller T5 models, but are also exhibited by models such as LLaMA when evaluated on MED-EASI.

## 6 Discussion

We discuss our results around the following aspects: the access level of the simplification models (open-vs. closed-weight), the training strategies (general

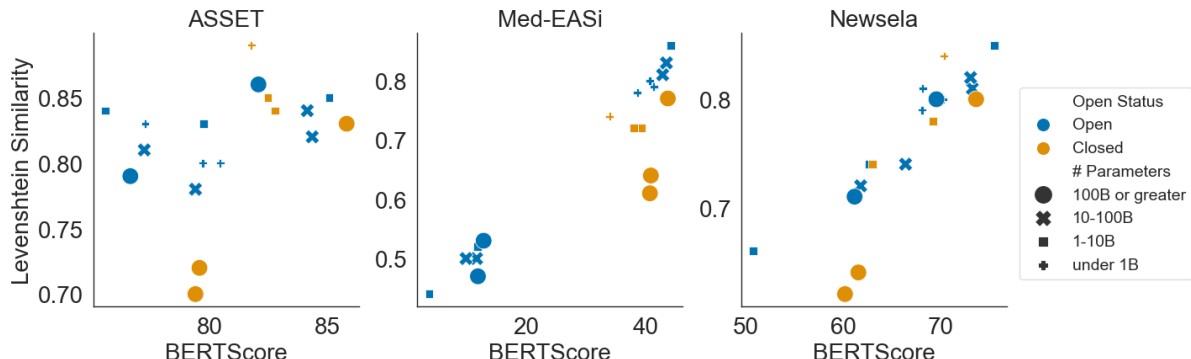

Figure 4: BERTScore, computed between the system output and reference sentence(s), correlates strongly with Levenshtein similarity, computed between the source sentence and system outputs. This indicates that BERTScore tends to reward minimally edited sentences. Levenshtein similarity is computed with the EASSE package (Alva-Manchego et al., 2019a).

pre-training vs. general fine-tuning strategies), and the utility of automatic metrics.

**Access Level**   Among the OpenAI models, we observe that all models perform particularly well on meaning preservation according to BERTScore but exhibit considerable differences in their ability to simplify, as indicated by SARI on 'weaker' models such as Ada-001. Among the evaluated open-weight models, we observe that the Flan models (T5 and UL2) typically perform competitively, punching well above their weight in terms of parameter counts with much larger decoder-only models. This is a promising finding for the category of open-weight models, and we hope that this encourages future work to continue investigating different methods regardless of the model size.

**Training Strategies**   Within model families, when comparing base models to their instruction fine-tuned counterparts, we observe that instruction-tuning typically leads to better performance in our few-shot ICL setting for TS. We find this to be particularly encouraging since TS is one task often hindered by the lack of high-quality labeled training data (Stajner, 2021).

Nevertheless, improvement is not always guaranteed, as seen when comparing BLOOM vs BLOOMZ. In this case, instruction fine-tuning leads to better-meaning preservation but a reduction in the degree of simplification, indicating that the instruction-tuning method used to derive the multilingual BLOOMZ may be less suitable for English TS. This stands in stark contrast to the Flan instruction tuning method, which delivers considerable gains in both SARI and BERTScore despite sharing the same underlying instruction-tuning dataset as

BLOOMZ. Therefore, we hypothesize that this drop in performance may be influenced by the multilingual instruction tuning setup that is unique to BLOOMZ.

**Utility of Automatic Metrics**   Overall, we find SARI and BERTScore to be useful automatic evaluation metrics for inspecting the trade-off between simplicity and meaning preservation (see Figure 6 in the Appendix). In general, closed-weight models often strike a more optimal balance. This is also supported by our qualitative analysis, which confirmed that these models rely less on deletion, an oft-overused operation (Devaraj et al., 2022), and more on other edits (e.g. paraphrasing or splitting).

Furthermore, our qualitative analysis shows that outputs with higher BERTScores tend to be minimally simplified, often copying the entire input text. We validate this by studying the relationship between BERTScore (computed between the system output and the reference sentence(s)) and Levenshtein similarity (computed between the system output and the original input sentence). Figure 4 reveals a strong positive correlation across all datasets, indicating that BERTScore tends to reward minimally simplified responses. For some of the closed-models, which tend to perform a greater degree of paraphrasing, this leads to lower BERTScores, while models that perform more copying are rewarded. Overall, the results from our qualitative study generally showed agreement with those from our automatic evaluation metrics, particularly SARI, BERTScore, and LENS. It also enabled us to pinpoint specific operations, such as re-ordering, and identify issues, notably hallucinations, in system outputs.

# 7 Conclusion

In this paper, we provided a comprehensive assessment of how well out-of-the-box LLMs perform on the task of TS with few-shot in-context learning. We found that the best LLMs outperform state-of-the-art supervised TS baselines while also producing a more diverse set of simplification operations. We also established that closed-weight models perform better than open-weight ones and that general instruction-tuning often improves a model's abilities on TS. Furthermore, we empirically validated the trade-off between simplicity and meaning preservation through automatic evaluation and a manual analysis. Our analyses of multiple few-shot prompting strategies revealed that a more structured prompting format produces better results than presenting source-target examples in continuous text.

Our performance benchmark, BLESS, provides a strong foundation for future work. For example, it remains an open question as to which expressions and instructions are optimal for prompting LLMs to simplify texts. Furthermore, this work exclusively focused on few-shot in-context learning. Future work could explore the capabilities of these systems in zero-shot, fine-tuned, or retrieval-based settings.

## Limitations

In this section, we discuss a few limitations of our work. First, we only considered English TS datasets, and it still remains to be seen how these TS abilities transfer to languages other than English. Additionally, we selected only a handful of output samples for manual analysis for the three test datasets considered, and all annotations were performed by one of the authors and subsequently validated by another author independently. It will be necessary to perform this at a larger scale to more accurately characterize the capabilities of each model for each domain and prompt. We further acknowledge the limits of the evaluation set itself. While we purposefully chose the test splits to cover a variety of domains, test splits for all three corpora amount to 915 samples, which could potentially limit the statistical power of results obtained from the assessment. Additionally, two out of the three test sets contain only sentences as input, while the third contains short multi-sentence texts, so this assessment mostly applies to the subtask of sentence simplification. Finally, our findings confirm that proprietary, closed-source models can achieve a new state-of-the-art performance on the task of text simplification. However, very little is known about their training data, alignment strategies, and implementation behind paywalled APIs. Therefore, the comparison to open-source models, which contain no explicit training on the task and an extremely bare-bones implementation is potentially unfair.

## Ethics Statement

This work is conducted in full awareness of and in line with the ACL Ethics Policy. Particularly, this work contributes to the transparency and fairness of evaluation methodologies in line with Sections 1.1, 1.2, 2.7, and 2.9 of the code, which innately leads to avoiding seen and unseen harms (Section 1.2, 1.4). We contribute to improving expertise in the domain of text simplification (Section 2.6). All models, datasets, and compute resources are used with permission and with concern to the appropriate access rights and licenses (Section 2.8). Our work contributes to the professional development of the research team (Section 3.5) and more widely benefits the research community and wider society (Section 3.1) by augmenting the understanding of the capacity of LLMs on the specific task of TS.

## Acknowledgements

We would like to thank Sian Gooding for her initiative in motivating this project, as well as Hoang Nguyen Hung Van, Jan Trienes, and everyone in the text simplification research community who joined our discussions during this journey. Thank you also to the anonymous reviewers for providing valuable feedback. This work was facilitated by the infrastructure services provided by S3IT, the Service and Support for Science IT team at the University of Zurich. Laura Vásquez-Rodríguez's work was funded by the Kilburn Scholarship from the University of Manchester.

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

# A  Model Details

In this section, we describe each type of LLM we use in our experiments.

## A.1  Open-weight Models

As a brief disclaimer, we note that some listed models are not truly "open-weight" and may require special permission to obtain weights for self-hosting. Further, in our descriptions, we do not distinguish between different variations of the same model. We provide the details of the training data and model sizes in Table 4. We consider both encoder-decoder and decoder-only models for our evaluation as discussed below.

### A.1.1  Encoder-Decoder Models

**T5 Family**  We evaluate a range of model variants derived from the original T5 models (Raffel et al., 2020). Originally, training recipes for T5 employ pre-training with a span-infilling objective and are thus not suitable for left-to-right generation tasks off the shelf. We thus use the T5-LM-adapted models from (Lester et al., 2021) which have undergone continued pre-training using a standard LM objective.

One later derivation includes the instruction-tuned variant Flan-T5 (Chung et al., 2022), which continues training from the aforementioned T5-LM-adapted checkpoints and uses a wide variety of labeled data for instruction fine-tuning. Notably, the dataset description by Chung et al. (2022) does not include any reference to simplification-related tasks. Similar parallel efforts lead to the creation of the T0 models (Sanh et al., 2022).

Finally, UL2 (Tay et al., 2023) proposes a more diverse set of pre-training objectives beyond simple span corruption. Additional tasks include sequence distortion and extreme span corruption.

### A.1.2  Decoder-only Models

**GPT-J/GPT-X**  Early reproduction efforts of large-scale GPT-style models started following the surge in popularity of GPT-2 (Radford et al., 2019). For our benchmark, we include models published by EleutherAI, namely the 6 billion parameter variant of GPT-J (Wang and Komatsuzaki, 2021) and the 20 billion parameter version of GPT-NeoX (Black et al., 2022). Both models were trained with a standard LM pre-training objective and were not fine-tuned to follow instructions.

| Models | Type | Sizes | Training Data |
|--------|------|-------|---------------|
| BLOOM | D | 560M, 1b1, 3b, 7b, 175b | ROOTS (Laurençon et al., 2022), Huggingface Datasets (Lhoest et al., 2021) |
| BLOOMZ | D | 560M, 1b1, 3b, 7b, 175b | P3 (Sanh et al., 2022), xP3 |
| LLaMA | D | 7b, 13b, 30b, 65b | CommonCrawl, C4 (Raffel et al., 2020), Github, Wikipedia, ArXiv, StackExchange |
| OPT | D | 1.3b, 6.7b, 13b, 30b, 66b | Pile (Gao et al., 2020), Reddit (Baumgartner et al., 2020) |
| OPT-IML | D | 1.3b, 30b, | OPT-IML Benchmark (Iyer et al., 2023) |
| GPT-J | D | 6b, 20b | Pile (Gao et al., 2020) |
| T5 | E-D | 60m (small), 220m (base), 770m (large), 3b (xl), 11b (xxl) | C4 (Raffel et al., 2020) |
| T0, T0pp | E-D | 3b, 11b | P3 (Sanh et al., 2022) |
| Flan-T5 | E-D | 60m (small), 220m (base), 770m (large), 3b (xl), 11b (xxl) | Muffin (Wei et al., 2022), P3 (Sanh et al., 2022), NIV2 (Wang et al., 2022) |
| UL2 | E-D | 20b | |
| Flan-UL2 | E-D | 20b | Muffin (Wei et al., 2022), P3 (Sanh et al., 2022), NIV2 (Wang et al., 2022) |

Table 4: Description of Open-Weight models. Model type "D" refers to decoder-only models, "E-D" for models based on an encoder-decoder architecture.

**OPT/LLaMA** Reproduction efforts of large-scale decoder-only models conducted by researchers at Meta AI were released under the OPT label (Zhang et al., 2022c) and more recently under the LLaMA label (Touvron et al., 2023). Besides a different composition in training data and some implementation choices relating to hardware performance, they otherwise share similar architectures and training objectives with the previously mentioned GPT-like models. Iyer et al. (2023) experimented with instruction tuning the OPT models to provide OPT-IML checkpoints, which we also use in BLESS.

**BLOOM** The result of an open collaboration, the BLOOM model family (Scao et al., 2022) represents the largest open-weight models available at the time of writing, up to the full 176 billion parameter scale of GPT-3 (Brown et al., 2020). The original model was only trained with a standard LM pre-training objective. BLOOMZ models (Muennighoff et al., 2022) extend these models with instruction fine-tuning.

### A.2 Closed-Weight Models

As the current primary choice for commercial solutions, we benchmark a range of models by OpenAI. Previous publications regarding the GPT family (Radford et al., 2018, 2019; Brown et al., 2020) establish that these models (Ada/Babbage/Curie/Davinci) are decoder-only, with varying numbers of parameters. Table 5 shows

the API inference costs of our experiments with OpenAI's models.

| Model | $/1k tokens | ASSET | MED-EASI | NEWSELA |
|-------|-------------|-------|----------|---------|
| Ada-001 | 0.0004 | 0.35 | 0.41 | 0.28 |
| Babbage-001 | 0.0005 | 0.44 | 0.51 | 0.35 |
| Curie-001 | 0.002 | 1.76 | 2.01 | 1.41 |
| GPT-3.5-Turbo | 0.002 | 1.75 | 1.95 | 1.37 |
| Davinci-002 | 0.02 | 17.62 | 20.06 | 14.10 |
| Davinci-003 | 0.02 | 17.52 | 19.90 | 13.96 |
| Total | – | 39.54 | 44.84 | 31.47 |

Table 5: Pricing information for OpenAI's API models. Here we report the total costs incurred for all three inference prompts and three seeded runs, totalling nine inference runs per dataset. Prices listed correspond to those for the API-based models available from April through June, 2023. All prices are in USD.

## B Supplemental Results

Tables 6, 7, and 8 show full results for these on ASSET, MED-EASI, and NEWSELA respectively.

### B.1 Details on Evaluation Metrics

A variety of automatic evaluation methods have been proposed. Commonly used automatic metrics like BLEU (Papineni et al., 2002) and SARI (Xu et al., 2016) can provide insights into how similar a model's outputs are to a set of gold reference simplifications. However, to more precisely understand a model's strengths and weaknesses, finer-grained evaluation is often required. For example, calculating the distribution of edit simplification operations (e.g. additions and deletions) (Vásquez-

Rodríguez et al., 2021a,b) can yield more insights into the capabilities of these systems. We evaluate model outputs according to multiple metrics. While we focus on reporting SARI and BERTScore in order to relate our findings with previous work, we also compute additional evaluation metrics for more fine-grained analyses and perform a qualitative analysis. Specifically, we report:

1. **SARI** (Xu et al., 2016): SARI (**S**ystem output **A**gainst **R**eferences and against the **I**nput sentence) is a holistic metric for simplification quality. It computes the F1 score for n-grams added, kept, and deleted, with respect to the input (source) and reference sentences.

2. **BERTScore** (Zhang et al., 2020): We compute the BERTScore precision, recall and F1 of the predictions against both the reference and source sentences, totaling in 6 different scores. Results reported in the paper use BERTScore F1 computed between system output simplifications and the gold reference sentence(s).

3. **FKGL** (Kincaid et al., 1975): FKGL (**F**lesch-**K**incaid **G**rade **L**evel) is a weighted score based on sentence length and syllable information. The lower the FKGL, the simpler the output, and the lowest possible score is -3.40. However, for a given test set, we consider the best FKGL to be the score that is closest to the FKGL of the gold references.

4. **LENS** (Maddela et al., 2023): LENS (**L**earnable **E**valuation **M**etric for **T**ext **S**implification) is a score between 0 and 100 estimated by a model trained on complex-simple pairs annotated with human ratings. We report the average LENS score for each dataset.

Table 6: Simplification Results on ASSET

| | SARI↑ | FKGL↓ | BERT↑ | LENS↑ |
|---|---|---|---|---|
| **Baselines** | | | | |
| Gold References | 45.27 | 6.53 | 78.89 | 65.58 |
| MUSS-mined | 42.29 | 8.18 | 79.86 | 61.36 |
| MUSS-wiki-mined | 44.90 | 5.29 | 77.71 | 69.23 |
| **LLMs** | | | | |
| Ada-001 | 33.97 | 9.06 | 81.76 | 56.41 |
| Babbage-001 | 38.44 | 8.65 | 82.46 | 61.39 |
| Curie-001 | 39.87 | 8.33 | 82.75 | 63.02 |
| Davinci-002 | 42.84 | 7.77 | **85.91** | 67.09 |
| Davinci-003 | 46.6 | 7.74 | 79.66 | 67.39 |
| GPT-3.5-Turbo | **47.69** | 7.51 | 79.39 | **69.17** |
| BLOOM-560m | 36.14 | 8.01 | 50.11 | 42.68 |
| BLOOM-1b1 | 34.08 | 8.18 | 68.60 | 51.23 |
| BLOOM-3b | 37.15 | 7.92 | 72.28 | 54.34 |
| BLOOM-7b1 | 36.96 | 8.17 | 77.82 | 57.37 |
| BLOOM | 39.72 | 7.78 | 76.63 | 60.37 |
| BLOOMZ-560m | 35.12 | 7.52 | 41.21 | 39.52 |
| BLOOMZ-1b1 | 35.00 | 8.42 | 76.66 | 54.86 |
| BLOOMZ-3b | 35.74 | 8.73 | 75.86 | 56.78 |
| BLOOMZ-7b1 | 37.05 | 8.56 | 79.09 | 59.14 |
| BLOOMZ | 37.63 | 8.27 | 82.06 | 61.07 |
| GPT-J-6b | 38.86 | 7.83 | 76.48 | 60.13 |
| GPT-NeoX-20b | 39.04 | 8.04 | 75.81 | 60.87 |
| LLaMA-7b | 40.70 | 7.39 | 75.52 | 62.80 |
| LLaMA-13b | 40.45 | 7.33 | 76.13 | 62.95 |
| LLaMA-30b | 39.14 | 7.32 | 78.74 | 62.73 |
| LLaMA-65b | 38.59 | 8.07 | 81.59 | 62.90 |
| OPT-1.3b | 33.01 | 8.61 | 75.57 | 57.08 |
| OPT-6.7b | 38.64 | 7.79 | 76.62 | 61.26 |
| OPT-13b | 38.78 | 8.03 | 79.08 | 60.51 |
| OPT-30b | 38.04 | 8.17 | 77.22 | 60.01 |
| OPT-66b | 39.64 | 7.76 | 76.72 | 61.68 |
| OPT-IML-Max-1.3b | 36.00 | 7.66 | 79.73 | 61.31 |
| OPT-IML-Max-30b | 42.03 | 6.62 | 79.39 | 65.29 |
| T0-3b | 35.16 | 8.90 | 54.92 | 50.38 |
| T0 | 36.49 | 8.56 | 55.32 | 48.71 |
| T0pp | 35.05 | 8.65 | 47.69 | 44.67 |
| T5-small-LM-adapt | 33.89 | **6.61** | 10.27 | 14.56 |
| T5-base-LM-adapt | 34.70 | 6.80 | 19.63 | 14.27 |
| T5-large-LM-adapt | 31.12 | 6.88 | 37.82 | 15.21 |
| T5-xl-LM-adapt | 29.12 | 7.06 | 48.25 | 23.39 |
| T5-xxl-LM-adapt | 33.17 | 6.85 | 46.59 | 25.43 |
| Flan-T5-small | 38.57 | 7.58 | 77.26 | 54.80 |
| Flan-T5-base | 41.40 | 7.32 | 79.70 | 62.75 |
| Flan-T5-large | 42.17 | 6.78 | 80.44 | 63.35 |
| Flan-T5-xl | 41.07 | 7.16 | 85.06 | 64.74 |
| Flan-T5-xxl | 41.75 | 7.27 | 84.13 | 66.08 |
| UL2 | 35.65 | 7.65 | 37.01 | 15.99 |
| Flan-UL2 | 42.83 | 6.85 | 84.34 | 67.36 |

## Table 7: Simplification Results on MED-EASI

| Model | SARI↑ | FKGL↓ | BERT↑ | LENS↑ |
|---|---|---|---|---|
| **Baselines** | | | | |
| Gold References | 100 | 9.59 | 100 | 65.89 |
| MUSS-mined | 35.15 | 9.29 | 42.55 | 52.48 |
| MUSS-wiki-mined | 35.12 | 8.04 | 43.07 | 59.12 |
| **LLMs** | | | | |
| Ada-001 | 36.52 | 10.62 | 33.95 | 41.43 |
| Babbage-001 | 36.60 | 10.49 | 37.95 | 53.91 |
| Curie-001 | 38.22 | 10.15 | 39.31 | 56.10 |
| Davinci-002 | 36.34 | 10.05 | 43.67 | 57.71 |
| Davinci-003 | 39.81 | **9.31** | 40.83 | 60.71 |
| GPT-3.5-Turbo | **40.14** | 8.93 | 40.67 | **63.80** |
| BLOOM-560m | 35.37 | 7.58 | -2.60 | 36.27 |
| BLOOM-1b1 | 35.86 | 7.37 | 1.63 | 40.47 |
| BLOOM-3b | 35.48 | 7.40 | 5.94 | 42.21 |
| BLOOM-7b1 | 37.47 | 7.23 | 9.53 | 44.17 |
| BLOOM | 37.72 | 7.11 | 11.95 | 47.50 |
| BLOOMZ-560m | 33.14 | 6.83 | -3.08 | 38.32 |
| BLOOMZ-1b1 | 35.65 | 6.99 | 6.40 | 43.69 |
| BLOOMZ-3b | 35.68 | 7.17 | 8.56 | 44.79 |
| BLOOMZ-7b1 | 36.78 | 7.08 | 9.43 | 47.15 |
| BLOOMZ | 36.60 | 7.08 | 12.90 | 47.67 |
| GPT-J-6b | 36.20 | 7.01 | 10.53 | 46.67 |
| GPT-NeoX-20b | 36.02 | 7.07 | 10.62 | 46.46 |
| LLaMA-7b | 36.95 | 6.62 | 10.28 | 48.42 |
| LLaMA-13b | 36.98 | 6.73 | 11.43 | 48.63 |
| LLaMA-30b | 37.56 | 6.89 | 12.21 | 47.92 |
| LLaMA-65b | 37.86 | 6.85 | 12.20 | 47.45 |
| OPT-1.3b | 34.00 | 7.17 | 3.82 | 43.64 |
| OPT-6.7b | 34.73 | 7.02 | 8.86 | 47.72 |
| OPT-13b | 34.69 | 6.96 | 8.73 | 47.16 |
| OPT-30b | 35.08 | 7.02 | 9.96 | 46.96 |
| OPT-66b | 35.72 | 6.96 | 11.42 | 47.28 |
| OPT-IML-Max-1.3b | 37.01 | 7.12 | 11.85 | 46.80 |
| OPT-IML-Max-30b | 35.80 | 6.78 | 11.73 | 49.28 |
| T0-3b | 38.16 | 10.34 | 17.83 | 42.02 |
| T0 | 35.67 | 10.81 | 15.93 | 42.76 |
| T0pp | 35.61 | 10.67 | 11.58 | 36.60 |
| T5-small-LM-adapt | 34.71 | 8.87 | -4.15 | 12.68 |
| T5-base-LM-adapt | 34.70 | 8.47 | -0.92 | 16.41 |
| T5-large-LM-adapt | 36.69 | 8.62 | 10.27 | 19.34 |
| T5-xl-LM-adapt | 33.65 | 8.83 | 18.91 | 22.59 |
| T5-xxl-LM-adapt | 32.61 | 9.10 | 21.69 | 28.21 |
| Flan-T5-small | 36.65 | 8.99 | 38.60 | 45.37 |
| Flan-T5-base | 36.79 | 9.05 | 40.63 | 51.95 |
| Flan-T5-large | 35.71 | 8.70 | 41.31 | 52.59 |
| Flan-T5-xl | 33.21 | 9.11 | **44.12** | 54.75 |
| Flan-T5-xxl | 34.27 | 9.13 | 43.43 | 54.70 |
| UL2 | 35.89 | 9.28 | 17.15 | 19.79 |
| Flan-UL2 | 35.31 | 8.52 | 42.80 | 57.95 |

## Table 8: Simplification Results on NEWSELA

| Model | SARI↑ | FKGL↓ | BERT↑ | LENS↑ |
|---|---|---|---|---|
| **Baselines** | | | | |
| Gold References | 60.11 | 5.88 | 87.66 | 71.02 |
| MUSS-mined | 38.40 | 7.86 | 72.14 | 61.49 |
| MUSS-wiki-mined | **41.24** | 6.12 | 74.10 | 67.61 |
| **LLMs** | | | | |
| Ada-001 | 34.42 | 8.66 | 70.33 | 55.06 |
| Babbage-001 | 36.41 | 8.32 | 62.99 | 60.91 |
| Curie-001 | 37.53 | 8.23 | 69.17 | 64.35 |
| Davinci-002 | 40.25 | 7.46 | 73.62 | **68.58** |
| Davinci-003 | 37.76 | 7.75 | 61.56 | 66.20 |
| GPT-3.5-Turbo | 37.29 | 7.80 | 60.19 | 67.97 |
| BLOOM-560m | 33.41 | 7.76 | 31.85 | 38.58 |
| BLOOM-1b1 | 35.37 | 7.99 | 48.52 | 46.54 |
| BLOOM-3b | 35.85 | 8.22 | 55.33 | 51.71 |
| BLOOM-7b1 | 36.12 | 7.96 | 61.00 | 54.16 |
| BLOOM | 37.48 | 7.49 | 61.17 | 60.98 |
| BLOOMZ-560m | 28.55 | 7.53 | 17.56 | 34.21 |
| BLOOMZ-1b1 | 35.22 | 7.47 | 54.19 | 53.05 |
| BLOOMZ-3b | 34.75 | 8.51 | 52.51 | 52.37 |
| BLOOMZ-7b1 | 36.21 | 8.29 | 59.53 | 59.36 |
| BLOOMZ | 37.06 | 8.41 | 69.55 | 62.07 |
| GPT-J-6b | 36.8 | 7.47 | 59.59 | 58.98 |
| GPT-NeoX-20b | 36.87 | 7.62 | 56.85 | 59.71 |
| LLaMA-7b | 36.70 | 6.28 | 55.31 | 62.43 |
| LLaMA-13b | 37.16 | 6.42 | 59.61 | 63.32 |
| LLaMA-30b | 37.50 | 6.75 | 63.89 | 64.30 |
| LLaMA-65b | 38.59 | 7.10 | 67.82 | 64.24 |
| OPT-1.3b | 34.76 | 7.96 | 50.78 | 55.35 |
| OPT-6.7b | 36.58 | 7.76 | 58.68 | 60.28 |
| OPT-13b | 37.67 | 7.16 | 60.65 | 61.31 |
| OPT-30b | 37.58 | 7.75 | 61.79 | 61.91 |
| OPT-66b | 37.45 | 7.25 | 60.43 | 62.98 |
| OPT-IML-Max-1.3b | 37.08 | 7.32 | 62.68 | 60.47 |
| OPT-IML-Max-30b | 39.59 | 6.09 | 66.39 | 64.74 |
| T0-3b | 33.37 | 8.50 | 36.56 | 50.64 |
| T0 | 32.83 | 7.58 | 30.96 | 53.23 |
| T0pp | 33.02 | 8.20 | 30.66 | 47.62 |
| T5-small-LM-adapt | 30.54 | 6.33 | 4.85 | 16.85 |
| T5-base-LM-adapt | 32.94 | **6.00** | 12.13 | 17.80 |
| T5-large-LM-adapt | 33.48 | 6.35 | 30.99 | 20.07 |
| T5-xl-LM-adapt | 32.85 | 6.62 | 42.51 | 25.48 |
| T5-xxl-LM-adapt | 33.44 | 6.67 | 44.15 | 29.49 |
| Flan-T5-small | 37.72 | 7.61 | 68.15 | 53.61 |
| Flan-T5-base | 38.67 | 7.21 | 68.09 | 59.11 |
| Flan-T5-large | 39.08 | 6.90 | 70.27 | 62.70 |
| Flan-T5-xl | 37.51 | 7.25 | **75.50** | 64.39 |
| Flan-T5-xxl | 39.42 | 7.32 | 73.05 | 65.13 |
| UL2 | 35.22 | 6.92 | 37.07 | 21.62 |
| Flan-UL2 | 40.27 | 6.86 | 73.23 | 66.42 |

| Model | Sentence | Annotation |
|---|---|---|
| Complex | They are rivaled as biological materials in toughness only by chitin. | - |
| Reference | They are rivaled only by chitin in toughness. | - |
| GPT-3.5-Turbo | Chitin is the only biological material that **rivals** them in **toughness**. | S↑ P+ L+ |
| Davinci-003 | Chitin is the only biological material **tougher** than them. | S↑ P+ L+ |
| Davinci-002 | They are tough like chitin, which is the **toughest** known biological material. | MP↓ P+ L+ |
| Flan-UL2 | They are only **second** to chitin for biological materials. | MP↓ P+ L+ |
| Flan-T5-large | Chitin is better than human materials in **toughness**. | MP↓ P+ L+ |

Table 9: Annotation examples from a SARI-based model ranking. S: Simplification, P: Paraphrasing, L: Lexical Simplification, and MP: meaning preservation. We highlight lexical simplification in **bold** and conflicts in meaning preservation in red.

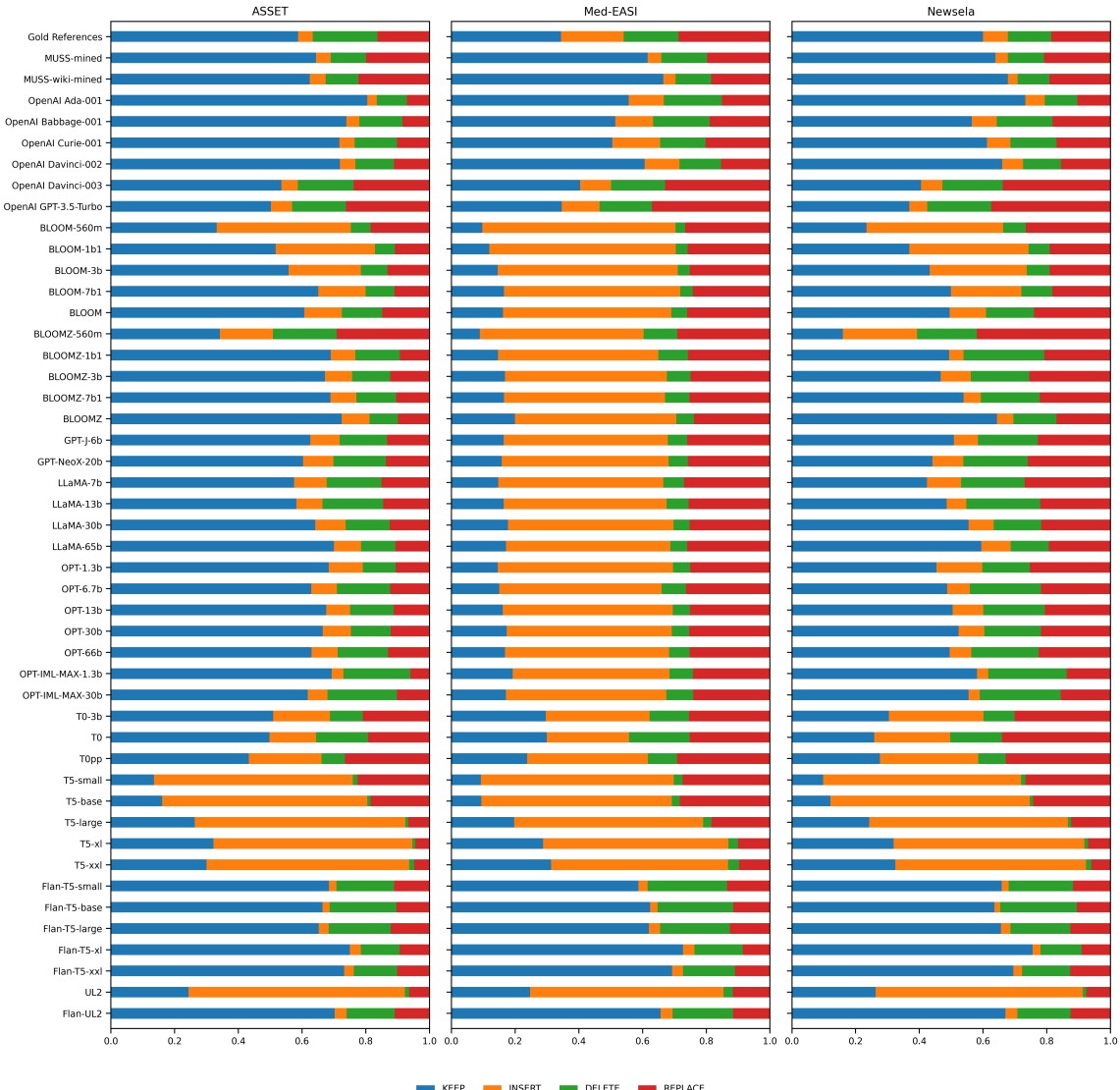

Figure 5: Token-level edit operations computed for all models and test sets using prompt 2. For most models, the edit operations performed in ASSET and NEWSELA reflect those in the gold reference simplifications. However, on the MED-EASI dataset, we observe a sudden spike in insertions from all LLMs except for OpenAI and Flan models. These additions indicate the presence of potentially unrelated hallucinated tokens and endless generations, which aligns with the low BERTScore results. We regard this failure case to be related to the fact that MED-EASI presents a challenging domain which is out of the distribution of most general-purpose models.

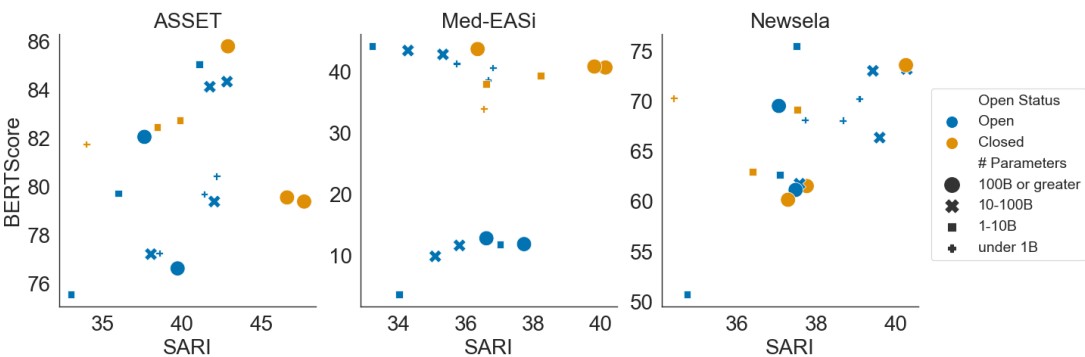

Figure 6: Adequacy-simplicity trade-off as exhibited by a limited set of models on each of the three datasets. On ASSET, higher SARI is associated with lower BERTScore. In the case of MED-EASI, we can see that smaller models, which often tend to copy the input sentence, are rewarded by BERTScore but punished by SARI. Here, only the closed-weight OpenAI models exhibit a favorable balance between the two metrics. On NEWSELA, the relationship is more linear. We suspect that this is influenced by the fact that reference sentences are taken from multiple simplification levels (1-4) and therefore cover a broader range of possible rewrites, some with more simplifying edit operations (rewarded by SARI) and some with fewer (rewarded by BERTScore).

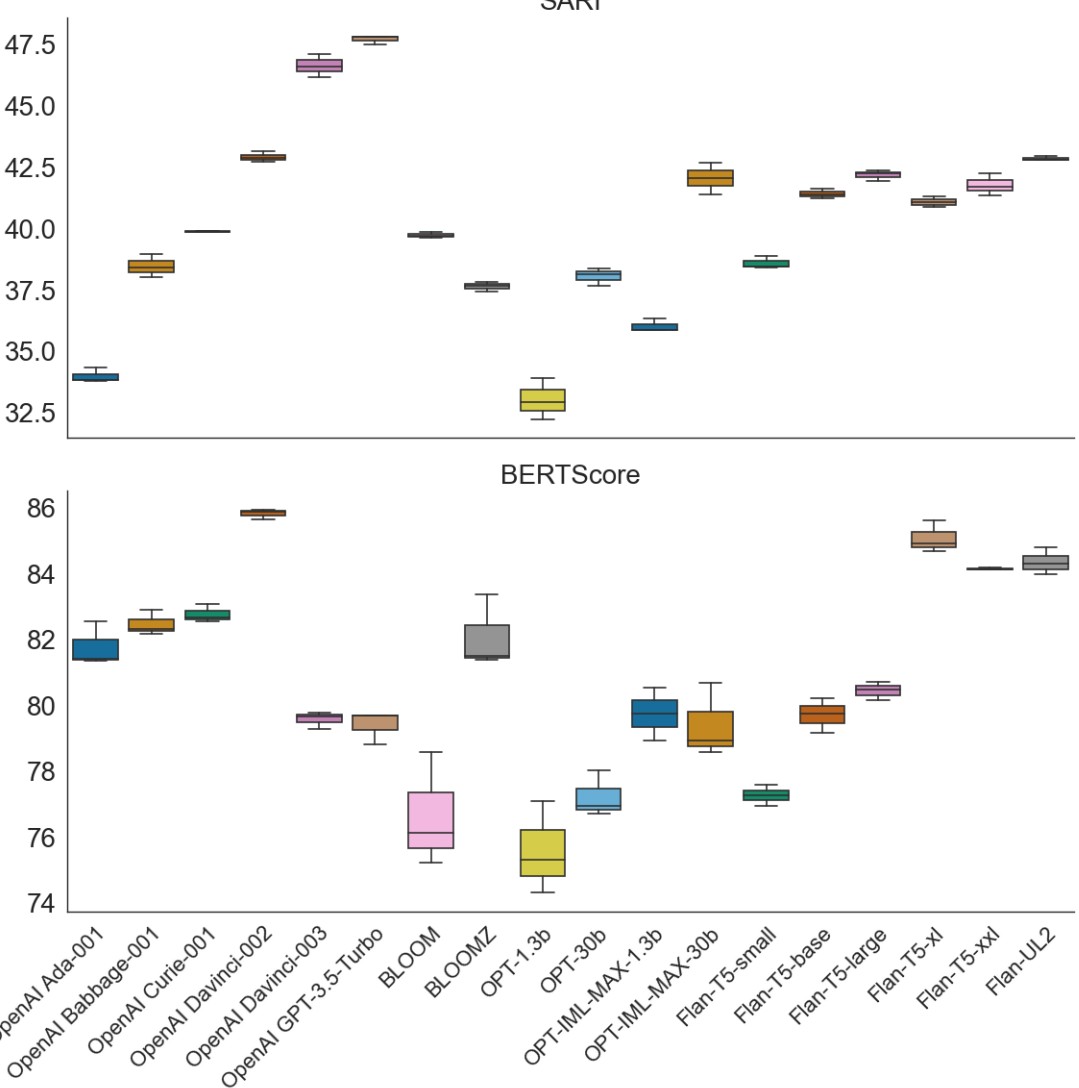

Figure 7: Visualizing LLM performance for select models, generated using prompt 2. This visualization corresponds to the results reported in Table 2 for ASSET. Models on the x-axis are ordered by model family, and within each model family, they are ordered by size (ascending).