# OpenReview forum: "BLESS: Benchmarking Large Language Models on Sentence Simplification"
_EMNLP/2023/Conference — EMNLP 2023 Main_

### Official Review · Reviewer_McU4 · 2023-08-03

**Soundness:** 3

**Excitement:**

3: Ambivalent: It has merits (e.g., it reports state-of-the-art results, the idea is nice), but there are key weaknesses (e.g., it describes incremental work), and it can significantly benefit from another round of revision. However, I won't object to accepting it if my co-reviewers champion it.

**Paper Topic And Main Contributions:**

The paper presents BLESS, a comprehensive performance benchmark of the most recent state-of-the-art Large Language Models (LLMs) on the task of text simplification (TS).

It evaluates 44 different LLMs on the TS task, including both open-source and commercial models with a range of sizes and architectures. The models are evaluated in a few-shot setting on 3 test datasets from different domains (Wikipedia, news, medical) using automatic metrics and analysis of simplification operations.

The study finds that closed-weight commercial models perform better than open-source models overall. Instruction tuning is shown to help improve meaning preservation.

There is a trade-off observed between simplicity and meaning preservation that varies across models and domains. The largest models tend to balance this trade-off better.

Analysis of simplification operations shows commercial models use more diverse operations compared to open models that overly rely on deletion.


**Reasons To Accept:**

1.	The paper makes an extensive evaluation of text simplification tasks using LLMs. It evaluates a large set of 44 diverse LLMs, covering different sizes, architectures, training methods, and access levels (open vs closed weights). This provides a comprehensive view of current LLM abilities.

2.	The authors combine automatic metrics, edit operations, and manual evaluation to thoroughly assess model performance from different angles. The qualitative analysis gives more nuanced insights.

3.	The paper provides a standardized benchmark that can serve as a strong baseline for future work and allow measuring progress in LLM-based simplification.

4.	The key results highlight the promise of LLMs for simplification but also identify limitations like inadequate domain generalization.

5.	The paper indicates which models may be most suitable for real-world applications today based on factors like accessibility, accuracy, training approach, etc.


**Reasons To Reject:**

Some potential weaknesses or limitations of this paper that could be improved on in future work:

1. It seems that this paper only evaluates multiple large language models for text simplification tasks, and is an extension of previous work, which is somewhat lacking in innovation.

2. The authors only evaluated datasets in three different domains and can try to extend the data to more domains and languages.
3. The test splits used to amount to 915 sentences total, which limits statistical power. Larger test sets could make results more robust. And, the manual evaluation only covers 300 samples, with just 5 outputs analyzed per model. A more extensive qualitative analysis could reveal more subtleties.
4. In addition to the quantitative analysis, the paper did not analyze and categorize the common failures. If some failure sample analysis can be carried out, it seems to be more able to show the need for improvement. Moreover, does not discuss broader implications or provide deeper insight into why models behave in certain ways.


**Reproducibility:**

3: Could reproduce the results with some difficulty. The settings of parameters are underspecified or subjectively determined; the training/evaluation data are not widely available.

**Reviewer Confidence:**

4: Quite sure. I tried to check the important points carefully. It's unlikely, though conceivable, that I missed something that should affect my ratings.

---

> ### Author Rebuttal · Authors · 2023-08-28
>
> Thank you for the time taken to read our paper and for providing valuable feedback. We have responded to each point of concern below.
>
> 1. The main contribution of our paper lies in the evaluation of 44 LLMs both quantitatively and qualitatively for text simplification, which is a task where these models have yet to be tested comprehensively. Explicitly, we contribute 1) an investigation of a few-shot approach to text simplification, 2) an extensive automatic evaluation of the system outputs, including the analysis of edit operations and 3) a qualitative analysis that include manual annotations and correlations between the selected evaluation metrics. Our findings also reveal GPT-3.5-Turbo attains state-of-the-art performance on ASSET and Med-EASI, which establishes important baselines for future work.
>
> 2. We agree that investigating more domains and languages would indeed be valuable. The domain-coverage for this study was limited by the availability of sentence-level simplification data. Our work represents the first iteration of the BLESS benchmark, and we hope that by publishing in the EMNLP conference, we will be able to build a community of Text Simplification researchers that contribute to BLESS by evaluating in the same manner, widening its scope in terms of datasets, languages and models. Since most open-weight LLMs are predominantly trained on English language data, we decided to focus solely on English in this work. Extending our approach to other domains and languages would be straightforward given appropriate test datasets and models with exposure to non-English data, and would also make a valuable contribution, which we encourage future authors to look into (as done in concurrent work (Ryan et al., 2023: https://arxiv.org/pdf/2305.15678.pdf).
>
> 3. We agree that the total number of test instances from the three datasets is a limiting factor. However, investigating 44 LLMs with 3 prompts and 3 random seeds on the 915 sentences amounted to roughly 360K model outputs. Despite the relatively small number of samples annotated as part of our qualitative analysis, we were still able to identify categorical problems that we can expect to find in larger sample sizes as well, even from the best performing open-weight models.
>
> 4. As part of our qualitative evaluation, we were able to identify some common failure cases (inappropriate changes to the meaning, ungrammatical outputs, and hallucinations (mentioned on lines 387-394). Rather than identifying specific failure cases for each model, we opted to measure these for each of the main metrics in order to compensate for the metrics’ blindspots. Table 3 provides a detailed summary of these phenomena.
>
> Regarding the reproducibility of our study, we intend to release our code with full details of inference parameters used and will elaborate on these in the final paper. While Newsela is indeed only available under a restrictive license, both ASSET and Med-EASI are publicly available. For these datasets, we plan to release the model outputs, which we think will be useful for future work on evaluation metrics.

---

### Official Review · Reviewer_2xuh · 2023-08-03

**Soundness:** 5

**Excitement:**

4: Strong: This paper deepens the understanding of some phenomenon or lowers the barriers to an existing research direction.

**Paper Topic And Main Contributions:**

The paper presents an extensive performance benchmark of 44 recent state-of-the-art Large Language Models on the task of Text Simplification (TS), which they call BLESS. The evaluation is based on three different datasets with original sentences/documents and their humanly written simplifications: ASSET (Wikipedia texts), MED-EASt (short medical texts), NEWSELA (professionally rewritten news article simplifications). The 44 models cover both open and closed sourced models, a range of different LLM architecture families, pretraining objectives and parameter sizes.

The models are tested applying few-shot in-context learning, so they are given a specific prompt describing the TS task together with 3 example complex--simplification pairs sampled from the datasets.

The authors evaluate all models on automatic evaluation metrics commonly used for TS, mostly SARI (measuring simplicity) and BERTScore (measuring content/meaning preservation), but also FKGL (also roughly measuring simplicity) and LENS (predictions by a model that was trained on human ratings for TS).
Additionally to these general metrics of simplification quality, they inspect the portion of operations that are performed by the models, using the Wagner-Fisher algorithm that detects insertion, replacement, deletion and keep operations.

In addition to the automatic evaluation (which are known to have weaknesses), the authors present a manual qualitative analysis of a selection of different model output simplifications (300 in total), where they rate the outputs considering e.g. if it is a valid simplification, mark common errors such as meaning change, ungrammatical text and hallucinations, marking the used operations (lexical simplification, deletion, sentence splitting, reordering, paraphrasing).

They further give some interesting secondary analyses, e.g. finding mostly a trade-off between simplicity and meaning preservation or showing high correlation of BERTScore and Levenshtein distance (showing that this measure might not be the best for TS evaluation, as simply copying the source text achieves high BERTScore).

Through these two evaluation approaches they achieve a number of interesting result, including:
* the best LLMs outperform standard TS baselines
* LLMs show a more diverse range of operations (deletions, paraphrases, lexical simplifications)
* the closed-weight models in general perform better than the open sourced ones
* in perliminary experiments they tried different versions of their TS prompt, noticing that highly structured example presentation works best
* larger models tend to be better at preserving meaning, but fail sometimes in terms of simplification
* models that were trained with instruction-tuning or RLHF (reinforcement learning with human feedback) have slightly better quality
* overall, all models performed best on the ASSET dataset, and worst on the medical dataset, showing the difficulty of the latter domain
* in general, the qualitative results are in-line with the automatic ones

**Questions For The Authors:**

1.) I could be wrong and might have overlooked but did you assigns some kind of overall "quality score(s)" in terms of simplicity / content preservation or TS quality in general in your manual evaluation? So sth. that would be directly comparable to BERTScore or SARI? I think that would be really interesting, both for the evaluation itself but of course also for assessing the usefulness of these automatic measures in comparison with human assessments. Maybe you could elaborate a bit on the description of Table 8 in the appendix.

**Reasons To Accept:**

I really enjoyed reading the paper and I would strongly argue to accept it! It has a number of interesting findings and overall is very complete and sound, as well as well-written and well-structured. It is very clear that the paper presents a very large work effort, which I really appreciate and deem helpful for others. Their choice of datasets from different domains, as well as the decision to use both automatic and manual evaluation methods is reasonable and fair.
The presentation (choice of figures/graphs, comparisons etc.) is well-thought through and makes it easy to follow the description of results, which in general, they present very well-structured. I also find the findings in the appendix very interesting.

**Reasons To Reject:**

I do not see any reasons to reject the paper.

**Reproducibility:**

3: Could reproduce the results with some difficulty. The settings of parameters are underspecified or subjectively determined; the training/evaluation data are not widely available.

**Reviewer Confidence:**

4: Quite sure. I tried to check the important points carefully. It's unlikely, though conceivable, that I missed something that should affect my ratings.

**Typos Grammar Style And Presentation Improvements:**

* page 4, line 313: do you mean Davinci-002 instead of Davinci-003?
* page 8, line 506: "trade-off"

---

> ### Author Rebuttal · Authors · 2023-08-28
>
> Thank you very much for taking the time to read our paper and for your feedback. We are glad to hear that you enjoyed reading our work. We have responded to the points raised below.
>
> 1.1 Single quality score: No, we did not assign a single score to the outputs assessed. While this would indeed be useful to relate to the automatic metrics, we believe single value scores are generally less useful for identifying specific strengths and weaknesses of different systems. Instead, we opted for assigning binary annotations to each output sentence. For example, S↑ means that the output was simplified. Although the percentage of outputs that received S↑ is not directly comparable with SARI, it is meant to convey the same thing - the simplicity of the model outputs. The same goes for MP↑, which conveys meaning preservation like BERTScore. The primary goal of the annotation scheme is to highlight the simplification capabilities of the models at first glance and how these are reflected given the automatic metrics. We agree that in future iterations of our work, it will be valuable to score our simplification based on larger sets of system outputs, as presented by previous work where they assess how well automatic metrics correlate with human assessments (see for example Alva-Manchego, et al., 2021 (https://direct.mit.edu/coli/article/47/4/861/106930/The-Un-Suitability-of-Automatic-Evaluation-Metrics), Maddela et al., 2022 (http://arxiv.org/abs/2212.09739)).
>
> 1.2 Table 8: In Table 8, we show an example of our annotation scheme over a sampled system output. We labeled the presence of a valid simplification as S↑, Paraphrasing as P↑ (rewriting a text without necessarily simplifying it), Lexical Simplification as L+ (replacing complex words or phrases with simpler ones), including highlighting this change in bold. We also highlight in red the lack of meaning preservation as MP↓. Since it’s possible that these annotations can overlap, it is difficult to reduce this to a single score.
>
> We thank you for your suggested grammatical edits and will incorporate those accordingly. Page 4 indeed refers to Davinci-002.
>
> Regarding the reproducibility of our work, our inference settings (especially with larger models) required a significant amount of GPU compute. That aside, two out of the three test datasets (ASSET and Med-Easi) should be reproducible with our code, which we will release upon acceptance. The Newsela dataset is available under a restrictive license but is also supported in our public code release.

---

### Official Review · Reviewer_WJrH · 2023-08-05

**Soundness:** 4

**Excitement:**

4: Strong: This paper deepens the understanding of some phenomenon or lowers the barriers to an existing research direction.

**Paper Topic And Main Contributions:**

This paper benchmarks 44 Large Language Models on the task of sentence simplification.

**Questions For The Authors:**

Question A: Can you provide an example of the inputs and outputs for a sentence simplification task?

Question B: Could you include the date on which you ran the experiments for closed-access models (since they are constantly being updated) and also provide pricing information for the sake of reproducibility?

**Reasons To Accept:**

The set of Large Language Models evaluated is extensive and one of the most comprehensive I've seen. The authors benchmarked 44 LLMs ranging in size from 60M to 176B parameters and include instruction-tuned models.

**Reasons To Reject:**

Though the authors do their own small analysis in Section 5, the bulk of this paper uses existing metrics to evaluate existing models on existing datasets.
The authors do not provide an example of sentence simplification, and it is unclear from their statement of the task (“rephrasing part or all of a sentence into language which is more widely understood“) does not distinguish the task from paraphrasing. -- after rebuttal period, I am satisfied with the authors' distinction between paraphrasing and sentence simplification

**Reproducibility:**

4: Could mostly reproduce the results, but there may be some variation because of sample variance or minor variations in their interpretation of the protocol or method.

**Reviewer Confidence:**

4: Quite sure. I tried to check the important points carefully. It's unlikely, though conceivable, that I missed something that should affect my ratings.

---

> ### Author Rebuttal · Authors · 2023-08-28
>
> Thank you for taking the time to read our paper and for providing useful feedback. Unfortunately, it seems that the text in “Reasons to Accept” is cut off. We would very much appreciate it if you could kindly complete this section.
>
> Below, we have summarised and responded to each point of concern from the review.
>
> - Existing models: Our primary goal with this paper was to understand how existing LLMs perform on the task of text simplification and provide a common evaluation benchmark. Therefore, we focused on evaluating as many LLMs as we could and did not train any new models.
> - Existing metrics and existing datasets: In order to directly compare our results with those from previous work, we chose to evaluate with widely used text simplification metrics and datasets. However, the goal of our qualitative analysis is to help fill gaps and diagnose problems with the simplification outputs that are difficult to detect with automatic metrics and likert-scale human ratings.
> - Description of the task: The comparison with paraphrasing is valid, since both tasks involve rewriting with the aim of retaining meaning. Paraphrasing’s only constraint is that the rewritten output text must retain the original text’s meaning. However, text simplification imposes an additional constraint, namely that the output should reduce the complexity of the input sentence and improve readability (e.g. avoiding rare or difficult words, reformulating challenging grammatical structures like cleft sentences into their constituent parts, reducing anaphora, deleting unimportant information, etc.). To address the point raised, we will elaborate on the task description and link it to section 4.2, which lists the general types of edit operations associated with simplification.
>
> Question A:
>
> Here is an example of a complex sentence with its simplification taken from the Newsela dataset:
>
> Complex: “A federally recognized tribe is considered a sovereign nation that has a government-to-government relationship with the United States and possesses the right to create and enforce laws, regulate activities within its jurisdiction and determine membership criteria.”
>
> Simple: “Recognized tribes are considered independent nations. They can make their own laws. They can govern their territory. Recognized tribes decide who is a member.”
>
> Question B:
>
> Experiments with the API-based models were run between late April and mid June, 2023.
> The total inference costs for these using OpenAI models was about $115 (USD). Below is a detailed breakdown of the inference costs per model/dataset for our particular experiments. Note that we ran inference 3 times for each different prompt on each dataset, totalling 9 runs.
>
> openai-gpt-3.5-turbo (USD 0.002 / 1k tokens):
> asset: USD 1.75, med-easi: USD 1.95, newsela: USD 1.37
>
> openai-text-ada-001 (USD 0.0004 / 1k tokens):
> asset: USD 0.35, med-easi: USD 0.41, newsela: USD 0.28
>
> openai-text-babbage-001 (USD 0.0005 / 1k tokens):
> asset: USD 0.44, med-easi: USD 0.51, newsela: USD 0.35
>
> openai-text-curie-001 (USD 0.002 / 1k tokens):
> asset: USD 1.76, med-easi: USD 2.01, newsela: USD 1.41
>
> openai-text-davinci-002 (USD 0.02 / 1k tokens):
> asset: USD 17.62, med-easi: USD 20.06, newsela: USD 14.10
>
> openai-text-davinci-003 (USD 0.02 / 1k tokens):
> asset: USD 17.52, med-easi: USD 19.90, newsela: USD 13.96
>
> We agree that these details are useful regarding reproducibility and will be sure to include them in the appendix.

---

### Meta-Review · Area_Chair_iuAR · 2023-09-18

**Recommendation:** 4

**Metareview:**

The paper presents an extensive performance benchmark of 44 recent state-of-the-art Large Language Models on the task of Text Simplification (TS), which they call BLESS. The evaluation is based on three different datasets with original sentences/documents and their humanly written simplifications: ASSET (Wikipedia texts), MED-EASt (short medical texts), NEWSELA (professionally rewritten news article simplifications). The 44 models cover both open and closed sourced models, a range of different LLM architecture families, pretraining objectives and parameter sizes.


Reasons To Accept:
- The set of Large Language Models evaluated is extensive. The authors benchmarked 44 LLMs ranging in size from 60M to 176B parameters and include instruction-tuned models.

Reasons To Reject:
- Though the authors do their own small analysis in Section 5, the bulk of this paper uses existing metrics to evaluate existing models on existing datasets.

The paper is a very interesting and enjoyable read and to me it seems very sound, detailed, comprehensive (they use 44 models and test on 3 different domain, using both automatic as well as manual/qualitative evaluation) and is well thought-through and structured and clearly written. The sheer number of models and evaluation methods makes it very apparent that furthermore they must have put a lot of effort in all the experiments. Moreover, the task of Text Simplification is interesting and relevant, as is - of course - the inspection of what LLMs really can do and cannot do.

---

### Decision · Program_Chairs · 2023-10-07

**Decision:**

Accept-Main

**Comment:**

The paper presents an extensive performance benchmark of 44 recent state-of-the-art Large Language Models on the task of Text Simplification (TS), which they call BLESS. The evaluation is based on three different datasets with original sentences/documents and their humanly written simplifications: ASSET (Wikipedia texts), MED-EASt (short medical texts), NEWSELA (professionally rewritten news article simplifications). The 44 models cover both open and closed sourced models, a range of different LLM architecture families, pretraining objectives and parameter sizes.


Reasons To Accept:
- The set of Large Language Models evaluated is extensive. The authors benchmarked 44 LLMs ranging in size from 60M to 176B parameters and include instruction-tuned models.

Reasons To Reject:
- Though the authors do their own small analysis in Section 5, the bulk of this paper uses existing metrics to evaluate existing models on existing datasets.

The paper is a very interesting and enjoyable read and to me it seems very sound, detailed, comprehensive (they use 44 models and test on 3 different domain, using both automatic as well as manual/qualitative evaluation) and is well thought-through and structured and clearly written. The sheer number of models and evaluation methods makes it very apparent that furthermore they must have put a lot of effort in all the experiments. Moreover, the task of Text Simplification is interesting and relevant, as is - of course - the inspection of what LLMs really can do and cannot do.